# Associations between spice or pepper (*Capsicum annuum*) consumption and diabetes or metabolic syndrome incidence

Azam Ildarabadi[1☯], Firoozeh Hosseini-Esfahani[1☯]*, Shahrzad Daei[1], Parvin Mirmiran[1,2]*, Fereidoun Azizi[3]

**1** Nutrition and Endocrine Research Center, Research Institute for Endocrine Sciences, Shahid Beheshti University of Medical Sciences, Tehran, Iran, **2** Department of Clinical Nutrition and Dietetics, Faculty of Nutrition Sciences and Food Technology, National Nutrition and Food Technology Research Institute, Shahid Beheshti University of Medical Sciences, Tehran, Iran, **3** Endocrine Research Center, Research Institute for Endocrine Sciences, Shahid Beheshti University of Medical Sciences, Tehran, Iran

☯ These authors contributed equally to this work.
* parvin.mirmiran@sbmu.ac.ir, Parvin.mirmiran@gmail.com (PM); f.hosseini@sbmu.ac.ir (FHE)

**Data Availability Statement:** The datasets generated and analyzed during the current study are not publicly available for ethical or legal reasons due to confidentiality agreements and data are

## Abstract

### Background

Spice and pepper are recognized as sources of antioxidants and anti-inflammatory compounds. This study investigated the association between spice or pepper intake and metabolic syndrome (MetS), related risk factors, or type 2 diabetes (T2D) incidences.

### Methods

The qualified Tehran Lipid and Glucose Study (TLGS) participants were included. In all examinations, dietary, anthropometrical, and biochemical variables were measured. Multivariable Cox proportional hazards regression models were exploited to determine the relationship between spice or pepper consumption and the hazard ratios for Type 2 Diabetes (T2D), Metabolic Syndrome (MetS), or its components.

### Results

The analysis was performed on 5340 individuals, with a mean age of 39.9±13.4 and 406 incident cases of T2D. Also, 4353 participants were included for MetS analysis with 1211 incident cases and a median follow-up of 5.8 years. After adjusting for confounding factors, spice and pepper intakes were not associated with T2D or MetS incidence. Further, in the upper quartile of spice intake, the HRs of high triglyceride (TG) [HR Q4: 1.19 (CI: 1.05–1.35)] and high blood pressure (BP) [HR Q4: 1.16 (CI: 1.04–1.30), P-trend = 0.007] increased. The risk of HDL-C appeared to decrease in the third quartile of pepper consumption (HR: 1, 0.97, 0.87, 1.03, P-trend = 0.008).

### Conclusion

The findings showed that spice and pepper consumption had no association with the incidence of T2D and MetS. The risk of high TG and high BP incidence was elevated in the

owned by Research Institute for Endocrine Sciences, Shahid Beheshti University of Medical Sciences. Tehran, Iran. Data are available from the ethics committee of the Research Institute for Endocrine Sciences, Shahid Beheshti University of Medical Sciences, Tehran, Iran, whenever data request has been sent. No.24, Arabi Street, Yemen Avenue, Chamran Highway Fax: +98 (21)22402463 Postal code: 1985717413 Email: info@endocrine. ac.ir.

**Funding:** This work was supported by the Research Institute for Endocrine Science, Shahid Beheshti University of Medical Science (Tehran, Iran) under Grant number. 43003670 the funders had no role in study design, data collection and analysis, decision to publish, or preparation of the manuscript.

**Competing interests:** The authors have declared that no competing interests exist.

upper quartiles of spice intake. Also, greater consumption of pepper decreased the incidence of low HDL-C.

## Introduction

Metabolic syndrome (MetS), as a prevalent condition, could result in non-communicable diseases (for instance, cardiovascular diseases, T2D, fatty liver disease) [1]. Various clinical manifestations like high waist circumference (WC), high blood pressure (BP), impaired fasting blood glucose (FBG), high level of triglyceride (TG), and high-density lipoprotein cholesterol (HDL-C) lower than normal levels are defined as MetS [2]. T2D corresponds to growing morbidity and mortality and is considered growing public health [3]. Also, it seems that the prevalence of T2D and obesity coincide with the frequency of MetS. While the MetS complications are increased, the management and prevention of this condition are raising interest [1]. Inflammation is a serious problem in patients with T2D, in which the circulating inflammatory cytokine levels and insulin resistance in the vascular endothelium, liver, and skeletal muscles increase [4, 5]. In this regard, spices and pepper are recognized as sources of antioxidants and anti-inflammatory compounds, so their use has gained attention in diabetes research [6]. Although modern medicine has made significant progress, using herbs and spices as complementary medicine has attracted much attention. Herbs and spices have compounds called polyphenols. Polyphenols have potentially beneficial effects on the body related to decreasing MetS [7]. Epidemiological and randomized clinical trial studies conducted on human and animal models have provided substantial evidence that dietary intake of phenolic compounds (PC) is vital in reducing the risk of diabetes. PC regulates significant pathways in carbohydrate metabolism including glycolysis, glycogenesis, and gluconeogenesis [8, 9]. Furthermore, PC has been found to have a direct inhibitory effect on the progress of inflammation and can quench appetite by inducing the liberation of cholecystokinin (CCK) and leptin; such action of PC may have an anti-obesogenic result and provide some level of protection against diabetes [9, 10].

In addition to anti-microbial, anti-inflammatories, and antioxidant properties, herbs and spices probably protect against cardiovascular disease, neurodegeneration, T2D, and cancer [11–13]. They also play an essential role in stabilizing lipid peroxidation and inhibiting various oxidizing enzymes [14]. Spices and culinary herbs also possess a high anti-glycans potential, mainly from their polyphenol content [15].

Previous epidemiological studies reported the controversial association of dietary polyphenols with reduced incidence of diabetes [16, 17]. Complementary medicine is used as an alternative therapy in various countries. Herbs and spices are considered to prevent and manage MetS due to the lack of side effects [18]; therefore, this study aims to evaluate the ingredients containing polyphenols in the usual Iranian foods. We decided to assess spices and pepper as a primary polyphenol that we added to prepare our foods.

Consequently, this cohort study investigated the associations between spice or pepper intakes and the incidences of T2D, MetS, or related risk factors.

## Materials and methods

### Study populations

Participants were chosen from the large-scale prospective TLGS, performed in the capital of Iran, Tehran, to identify non-communicable disease risk factors. The baseline survey, utilizing

the composite arranged bunch random sampling technique, was conducted on 15005 individuals aged ≥3 years from 1999 to 2001. The follow-up surveys were performed every three years; 2002–2005 (survey 2), 2005–2008 (survey 3), 2008–2011 (survey 4), 2012–2015 (survey 5), and 2015–2018 (survey 6) to recognize newly evolved risk factors or ailments.

Of 7898 subjects with complete dietary assessment in survey 4, 6882 were aged ≥18 years. Subjects reporting an energy intake of lower than 800 kcal/day or more than 4200 kcal/day were set aside from the study (n = 470) [19]. 6412 adult men and women remained as baseline participants and followed up until survey 6. Women who are pregnant or lactating and those diagnosed with T2D according to FBG evaluation or using blood glucose-lowering agents were excluded. (n = 523). Also, subjects without biochemical and anthropometric data (n = 84) were excluded. Subjects with missing data or missing follow-up data (n = 465) were excluded. Finally, 5340 participants remained and were involved in the secondary analysis of the association between spice or pepper intakes and the incidence of T2D (Fig 1).

Subjects participating in survey 4 (baseline) aged ≥18 years who completed dietary data were included for the secondary analysis of the association between spice or pepper consumption and MetS incidence. Those diagnosed with Mets [20], pregnant or lactating women, and individuals who take under 800 or over 4200 kcal/day of energy intake were excluded [19]. All subjects (n = 4,517), adult women and men, with attainable biochemical, anthropometric, and dietary information were included as the baseline community and monitored until Survey 6. Individuals lacking follow-up data were excluded from the analysis. Finally, 4353 subjects were imported in the analysis. Independent exclusions were also done for MetS components such as high BP, abdominal obesity, low HDL-C, high FBG, and high TG (Fig 2).

All participants prepared informed consent to take part in this study. This study received approval from the ethics committee of the Research Institute for Endocrine Sciences at Shahid Beheshti University of Medical Sciences (Tehran, Iran) [21, 22] (IR.SBMU.ENDOCRINE. REC.1401.118) (Grant no. 43003670).

## Dietary intake evaluation

Expert nutritionists conducted the dietary intake assessment utilizing a valid and reliable 147-item semi-quantitative food frequency questionnaire (FFQ) [23, 24]. In checking the validity and reliability of the questionnaire, peppers were examined with other vegetables as a group; however, the validity and reliability of spice intake were not evaluated.

Usual dietary intakes were obtained through facing personal interviews utilizing standard portion sizes. The FFQ questionnaire questioned the usual spice intake as a food item. The interviewer asked to report the usual spice intake as teaspoons/day and transformed to gram/ day. For green and bell peppers, the interviewer asked to report how many they use, and then they transformed intakes to grams in a day. Daily intakes were calculated by the frequency of intake for each food item whether daily, weekly, monthly, or yearly. Portion sizes were then modified to mass (gr/day) from the household measurements. The energy and nutrient intakes are determined by using the USDA (United States Department of Agriculture) Food Composition Table (FCT). The Iranian FCT provided national foods not included in the USDA [25]. Nutrient and food items are cumulatively calculated beyond follow-up examinations from the baseline until the final follow-up appointment or once the diabetes, MetS, or related risk factors appear.

## Physical activity measurements

Physical activity was computed utilizing a Persian-translated Modifiable Activity Questionnaire (MAQ). Previous research indicated that the MAQ has moderate validity and high

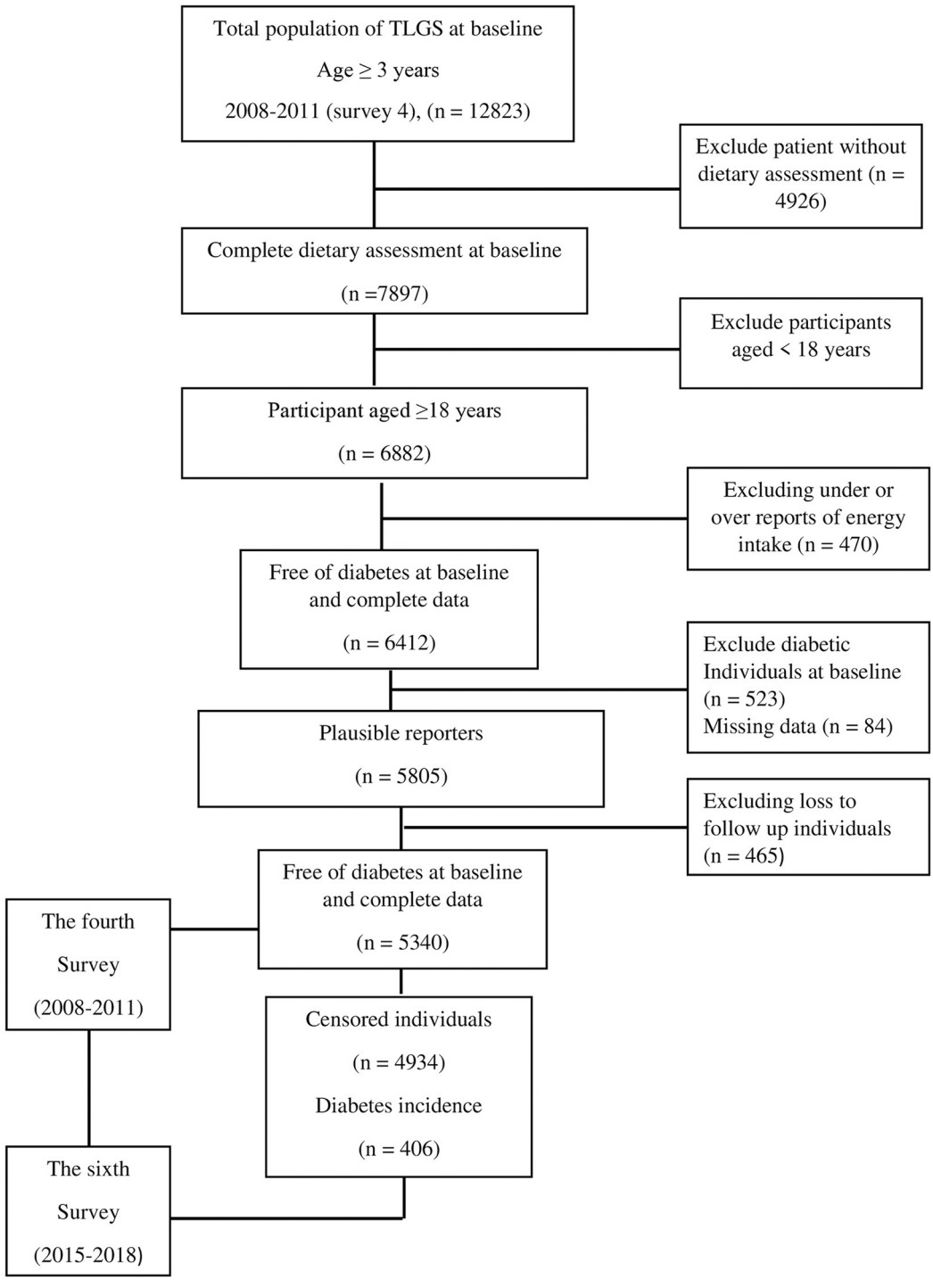

**Fig 1. Participant recruitment process flowchart.**

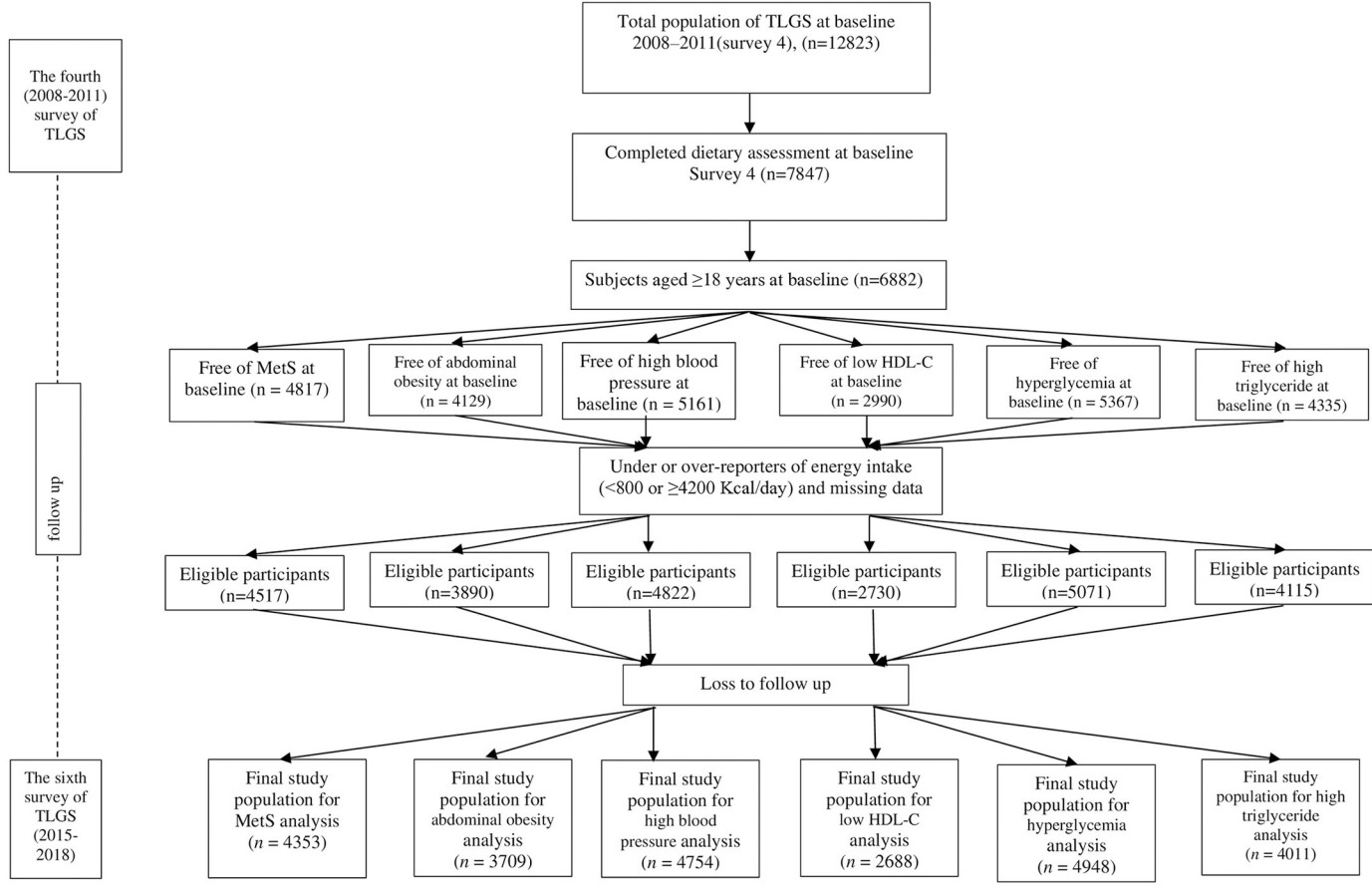

**Fig 2. Participant recruitment process flowchart.**

reliability [26]. The frequency and duration of light, moderate, high, and strenuous activities were evaluated depending on participants' typical daily routines over the past year. The intensity of routine daily activity (low, mild, moderate, severe), time, and frequency were measured beyond the past year. The daily activity data were transformed into metabolic equivalent/minutes/week (MET/min/week) [27].

## Blood pressure and anthropometric assessment

Body weight was evaluated with an accuracy of 100 gr utilizing a digital scale (Seca 707) with little wearing. The height was scaled with an accuracy of 0.5 cm with a tape meter on the wall in a straight and standing posture without shoes. WC and hip circumference (HC) were scaled exploiting a flexible tape meter with an accuracy of 0.1 cm. The HC was evaluated around the hips at the point where the width is greatest (above the gluteal fold). WC was assessed with an accuracy of 0.5 cm by holding the tape measure between the top of the hipbone and the bottom of the ribs. The waist-to-hip ratio was computed by dividing WC to HC.

BP was assessed using a standard mercury sphygmomanometer based on standard protocols that were previously explained [21]. After 15 minutes of rest, the physician fixed the cuff on the participant's right arm, and BP was measured while the participants were seated. After 30 seconds, the blood pressure measurement was repeated. The median blood pressure measurements were considered.

## Biochemical analysis

Samples of blood tests were gathered between 7 and 9 a.m. following a 12 to 14-hour fasting period. After 30 to 45 minutes of sample gathering, blood samples were centrifuged. All biochemical analyses were conducted on the same day in the TLGS research laboratory using the Selectra 2 auto-analyzer (Vital Scientific, Spankeren, the Netherlands). Fasting blood glucose (FBG) levels were measured using the enzymatic colorimetric and glucose oxidase techniques. During follow-up, for subjects who did not utilize blood glucose-lowering agents, the standard 2-h post-challenge blood glucose test was measured utilizing an oral prescription of 82.5 g glucose monohydrate solution (equivalent to 75 g anhydrous glucose). After precipitating apolipoprotein-B containing lipoproteins with phosphotungstic acid, the concentration of HDL-C was determined. Triglyceride (TG) levels were evaluated using enzymatic colorimetric tests that employed glycerol phosphate oxidase (Pars Azmoon Inc., Tehran, Iran). The inter-assay and intra-assay coefficients of variation for glucose were both 2.2%. For triglycerides, the inter-assay coefficient of variation was 1.6%, while the intra-assay coefficient of variation was 0.6% [21].

## Diabetes risk score

For reduction confounding variables number in the analysis, the diabetes risk score (DRS) was computed as mentioned variables: family history of T2D (5 points) (a positive family history of T2D was considered when at least one parent or sibling having T2D), systolic BP (mmHg) <120 (0 points), 120–140 (3 points), systolic BP ≥140 (7 points); TG/HDL-C: <3.5 (0 points), ≥3.5 (3 points); WC to-height ratio: <0.54 (0 points), 0.54–0.59 (6 points), ≥0.59 (11 points); FBG (mmol/L): <5.0 (0 points), 5.0–5.5 (12 points), 5.6–6.9 (33 points) [28].

## Outcome definition

When FBG concentrations ≥126 mg/dl or 2-h plasma glucose concentrations ≥200 mg/dl or self-declared consuming blood sugar-reducing agents (oral diabetes medication or insulin injections) appeared, T2D was detected [29]. If subjects exhibited three or more of the following criteria based on the Iranian-modified National Cholesterol Education Program/Adult [20, 30], Metabolic Syndrome (MetS) phenotype would has been detected: (1) blood pressure ≥130/85 mmHg or using the antihypertensive drug; (2) abdominal obesity (waist circumference ≥ 95 cm for both men and women); (3) HDL cholesterol levels <1.30 mmol/L (<50 mg/dL) in women and <1.04 mmol/L (<40 mg/dL) in men, or being treated with medication; (4) triglyceride levels ≥1.70 mmol/L (≥150 mg/dL) or therapeutic intervention; (5) fasting blood glucose (FBG) ≥6.11 mmol/L (≥110 mg/dL) or intervention for hyperglycemia.

## Statistical analysis

The SPSS for Windows, version 26 (IBM Inc.) was exploited for analyzing the data. A P-value of less than 0.05 was accepted as a remarkable change. We categorized the data on spice and pepper intake into quartiles. The Chi-square test for qualitative variables and one-way ANOVA for quantitative variables were utilized to assess the average and frequency of baseline features of subjects across quartiles of spice and pepper intakes. P for trends across spice and pepper intake categories was accomplished by appointing quantitative variables in a linear regression model.

The hazard ratios (HRs) and 95% confidence intervals (CIs) for the incidence of Type 2 Diabetes (T2D) and Metabolic Syndrome (MetS) or related risk factors were estimated using multivariable Cox proportional hazards regression analyses. No interactions were observed

between spice or pepper intake and age or sex groups. The first quartile was used as the reference group. The median of each quartile was used as a quantitative variable to calculate the P-value for trends in spice and pepper intake across the quartiles, utilizing Cox proportional hazards regression models. The time to event was determined as the period from baseline to either the last follow-up (for censored participants) or the event date (for those who experienced the event), whichever was recognized first. The event date was identified as the midpoint between follow-up visits leading to the first diagnosis of Type 2 Diabetes (T2D), Metabolic Syndrome (MetS), or its components, and the most recent follow-up before the diagnosis. Individuals were excluded from the analysis due to death or loss to follow-up. In the analysis for T2D incidence, sex, age, education levels (<8, 9–12, and >12 years), smoking (non-smoker, past smoker, and recent smoker), physical activity, DRS, total energy, total fat (percentage of energy), and fiber intake (gr/1000 kcal) were considered as cofounders in the adjusted model. The Cox regression models were executed with sex and age adjustments in the crude model. In the analysis for MetS incidence, age, sex, education level, physical activity, total energy, saturated fat (percentage of energy), fiber intake (gr/1000 kcal), BMI, and smoking were considered cofounders in the adjusted model. To estimate HR in models, high FBG, high BP, high TG, and low HDL-C were included as continuous risk factors at the beginning of the study, which were added to the adjustment variables. Further, variables with non-normally distributed shape were converted into natural log (Ln) variables to align with normal distribution before entering the statistical models. The proportional hazards assumption was confirmed by exploiting the Schoenfeld residuals test and a plot of log[-log(survival)] versus log(time) to verify their parallelism.

## Results

This study was executed on 2303 men and 3037 women (n = 5340) with the mean±SD age of women 39.0±13 and men 41.1±14; during an average follow-up of 5.8 years, there were 406 incident cases of T2D. For MetS analysis, there were 1694 men and 2659 women (n = 4353), with the mean±SD age of 38.6±14.3 and 35.9±11.8 years, respectively. During 5.8 years of follow-up, 1211 subjects were found with MetS incidence. The characteristics of participants at baseline and their dietary intakes among quartiles of spice intake are demonstrated in Tables 1 and 2, respectively. Results have shown that recent smokers were greater in the upper quartile of spice intake (Table 1). Also, WC and WC to height ratio (WHR) were lower in the upper quartile of spice intake. Furthermore, individuals in the upper quartiles of spice consumption had lower systolic and diastolic BP than subjects in the primary quartiles (P<0.001). Also, participants in the upper quartile of spice intake had lower TG/HDL-C ratio and FBG.

Participants in the upper quartile of spice intake demonstrated that a decrease in carbohydrate and protein intake (Table 2), while there was an increase in total fat, polyunsaturated fatty acids (PUFAs), monounsaturated fatty acids (MUFAs), saturated fatty acids (SFAs), and energy intake. Fiber consumption also increased in the upper quartile of spice intake. The average intake of spice is 1.6±1.4 grams per day, and the mean intake of pepper is 10.1±13.5 grams per day.

The HRs for T2D based on quartiles of spice and pepper intakes were shown respectively in Tables 3 and 4. larger intake of spice was related to a 1% elevation in the incidence of T2D (HR: 1, 0.99 (0.92–1.0), 1.0 (0.94–1.1), 1.1 (1.0–1.1), P for trend = 0.01); however, after adjusting for confounding factors, there was no relation between spice consumption and T2D incidence. Furthermore, the consumption of pepper had no association with incidence of T2D [HR: 1, 0.97 (0.90–1.0), 0.96 (0.89–1.0), 0.95 (0.88–1.0), P trend = 0.8]; also, this relation did

**Table 1. The characteristics of adult participants across quartiles of spice intake at baseline of TLGS.**

| Variables | Q1 | Q2 | Q3 | Q4 | P-value |
|---|---|---|---|---|---|
| **Mean ± SD (gr/day)** | 0.26±0.16 | 1.01±0.3 | 1.93±0.28 | 3.35±1.98 | |
| Age (years) | 42.8±14.3 | 39.4 ±13.0 | 39.2 ± 13.0 | 38.2±13.1 | <0.001 |
| Sex (% women) | 42 | 51.1 | 60.0 | 74.1 | <0.001 |
| Smoking n (%) | | | | | |
| Non-smoker | 902 (68.2) | 820 (63.0) | 881 (70.0) | 894 (67.2) | 0.01 |
| Ex-smoker | 250 (18.9) | 270 (20.7) | 238(18.1) | 220 (16.5) | |
| Smoker | 170 (12.9) | 212 (16.3) | 198 (15.0) | 216 (16.2) | |
| Physical activity (MET/min/week) | 520 ± 757 | 538 ± 896 | 532 ± 811 | 499 ± 758 | 0.60 |
| BMI (kg/m$^2$) | 27.1 ± 4.63 | 27.0 ± 4.72 | 27.0 ± 4.69 | 27.1 ± 4.93 | 0.80 |
| WC (cm) | 93.0 ± 11.7 | 91.8 ± 12.0 | 90.8 ± 12.0 | 90.2 ± 12.4 | <0.001 |
| WHR | 0.92 ± 0.07 | 0.91 ± 0.07 | 0.90 ± 0.07 | 0.90 ± 0.07 | <0.001 |
| Systolic BP (mm Hg) | 114 ± 16.2 | 112 ± 15.2 | 111± 15.2 | 110 ± 15.4 | <0.001 |
| Diastolic BP (mm Hg) | 76.7 ± 10.8 | 75.2 ± 10.3 | 74.6 ± 10.4 | 74.7 ± 10.7 | <0.001 |
| Total cholesterol (mg/dl) | 186 ± 38.8 | 184 ± 37.5 | 184 ± 37.1 | 183 ± 38.2 | 0.20 |
| TG/HDL-C ratio | 3.47 ± 3.08 | 3.16 ± 2.40 | 3.18 ± 2.70 | 2.84 ± 2.14 | <0.001 |
| FBG (mg/dl) | 93.3 ± 8.84 | 92.6 ± 8.57 | 92.7 ± 8.45 | 91.7 ± 8.33 | <0.001 |
| Family history of diabetes n(%Yes) | 124 (9.3%) | 109 (8.3%) | 135 (10.1%) | 151 (11.2%) | 0.31 |
| Education | | | | | |
| Elementary | 200 (15%) | 139 (10.6%) | 154 (11.5%) | 151 (11.2%) | <0.001 |
| Diploma | 759 (57%) | 764 (58.1%) | 809 (60.5%) | 849 (60.1%) | |
| Academic | 372 (27.9%) | 411 (31.3%) | 375 (28%) | 345 (25.7%) | |

Values are mean ± SD unless otherwise listed. P values were derived from the analysis of variance and Chi-square test for continuous and dichotomous variables, respectively.

BMI: body mass index, MET: Metabolic equivalent, WC: waist circumference, WHR: waist to height ratio, BP: blood pressure, TG: triglyceride, HDL-C: high-density lipoprotein ratio, FBG: fasting blood glucose.

[a]: An individual who has given up cigarette and/or tobacco smoking.

**Table 2. Dietary intake of adult subjects beyond quartiles of spice intake of TLGS.**

| Variables | Q1 | Q2 | Q3 | Q4 | P-value |
|---|---|---|---|---|---|
| **Mean ± SD (gr/day)** | 0.26±0.16 | 1.01±0.3 | 1.93±0.28 | 3.35±1.98 | |
| Carbohydrate (% of energy) | 60.2 ± 5.96 | 59.3 ± 5.33 | 59.1 ± 5.08 | 57.9 ± 5.47 | <0.001 |
| Protein (% of energy) | 15.1 ± 3.68 | 15.0 ± 2.49 | 14.9 ± 2.49 | 14.7 ± 2.81 | 0.002 |
| Total fat (% of energy) | 28.3 ± 7.80 | 29.1 ± 4.73 | 29.4 ± 4.65 | 31.0 ±5.18 | <0.001 |
| SFA (% of energy) | 10.2 ± 24.5 | 9.73 ± 2.70 | 9.82 ± 2.89 | 10.1 ± 2.76 | 0.005 |
| PUFA (% of energy) | 5.80 ± 6.29 | 5.86 ± 1.53 | 6.02± 1.45 | 6.28 ± 1.66 | <0.001 |
| MUFA (% of energy) | 10.1 ± 24.5 | 9.74 ± 2.37 | 10.1 ± 3.08 | 10.5 ± 2.64 | <0.001 |
| Fiber (gr/1000 kcal) | 9.99 ± 2.92 | 10.0 ± 2.75 | 10.5 ± 2.82 | 10.5 ± 3.33 | <0.001 |
| Energy intake (kcal/day) | 2172 ± 652 | 2330 ± 694 | 2416 ± 711 | 2494 ± 637 | <0.001 |

Data are presented as Mean ± SD. The ANOVA test was used, and the P-value for trends across the quartiles of spice consumption was calculated by incorporating continuous variables into a linear regression model.

SFA: saturated fatty acid, MUFA: Mono-unsaturated fatty acid, PUFA: poly-unsaturated fatty acid.

**Table 3. HRs (95% CI) of T2D, MetS, and its components incidence across categories of spice intake in adult subjects of TLGS.**

| Variables | | Q1 | Q2 | Q3 | Q4 | P-trend |
|---|---|---|---|---|---|---|
| | | **Quartiles of Spice intake** | | | | |
| **Incidence of MetS (n = 4353)** | | 328 | 316 | 288 | 279 | |
| **Median** (gr/day) (Min-Max) | | 0.26 (0.0–0.53) | 1.0 (0.53–1.5) | 2.0 (1.5–2.7) | 3.0 (2.7–33.1) | |
| HR (95% CI) | [a] Crude | Ref. | 0.98 (0.90–1.07) | 0.94 (0.87–1.03) | 1.08 (0.99–1.18) | 0.63 |
| | [b] Model adjusted | Ref. | 1.01 (0.91–1.12) | 1.02 (0.92–1.14) | 1.05 (0.94–1.18) | 0.65 |
| **Incidence of Low HDL-C (n = 2688)** | | 267 | 218 | 237 | 245 | |
| **Median** ((gr/day) | | 0.21 | 1.02 | 1.92 | 3.0 | |
| HR (95% CI) | [a] Crude | Ref. | 0.88 (0.79–0.98) | 0.88 (0.78–0.98) | 0.97 (0.87–1.09) | 0.23 |
| | [3] Model adjusted | Ref. | 0.95 (0.83–1.07) | 0.94 (0.81–1.07) | 0.97 (0.83–1.12) | 0.91 |
| **Incidence of High FBG (n = 4949)** | | 348 | 346 | 312 | 301 | |
| **Median** (gr/day) | | 0.26 | 1.03 | 1.99 | 3.0 | |
| HR (95% CI) | [a] Crude | Ref. | 0.97 (0.90–1.05) | 0.95 (0.88–1.03) | 1.08 (1.00–1.17) | 0.20 |
| | [3] Model adjusted | Ref. | 0.94 (0.85–1.04) | 0.95 (0.86–1.05) | 1.11 (0.99–1.24) | 0.08 |
| **Incidence of high TG (n = 4011)** | | 336 | 330 | 317 | 323 | |
| **Median** (Spice) (gr/day) | | 0.26 | 1.07 | 2.0 | 3.0 | |
| HR (95% CI) | [a] Crude | Ref. | 1.01 (0.93–1.11) | 0.96 (0.88–1.05) | 1.10 (1.01–1.21) | 0.39 |
| | [3] Model adjusted | Ref. | 1.05 (0.94–1.17) | 1.09 (0.97–1.23) | 1.19 (1.05–1.35) | 0.95 |
| **Incidence of High WC (n = 3709)** | | 248 | 259 | 237 | 237 | |
| **Median** (gr/day) | | 0.27 | 1.12 | 2.01 | 3.00 | |
| HR (95% CI) | [a] Crude | Ref. | 0.96 (0.88–1.06) | 0.91 (0.83–1.00) | 1.00 (0.91–1.10) | 0.83 |
| | [c] Model adjusted | Ref. | 0.96 (0.85–1.07) | 0.93 (0.83–1.04) | 0.98 (0.86–1.11) | 0.98 |
| **Incidence High BP (n = 4754)** | | 335 | 312 | 330 | 318 | |
| **Median** (gr/day) | | 0.26 | 1.07 | 2.00 | 3.00 | |
| HR (95% CI) | [a] Crude | Ref. | 0.98 (0.90–1.06) | 1.00 (0.92–1.08) | 1.16 (1.07–1.26) | 0.002 |
| | [c] Model adjusted | Ref. | 1.03 (0.93–1.14) | 1.06 (0.96–1.17) | 1.16 (1.04–1.30) | 0.007 |
| **Incidence T2D (n)** | | 101 | 99 | 98 | 109 | |
| **Median** (gr/day) (Min-Max) | | 0.26 (0.0–0.5) | 0.96 (0.5–1.5) | 1.8 (1.5–2.5) | 3.0 (2.5–43.2) | |
| HR (95% CI) | [a] Crude | Ref. | 0.99 (0.92–1.0) | 1.0 (0.94–1.1) | 1.1 (1.0–1.1) | 0.01 |
| | [d] Model adjusted | Ref. | 1.0 (0.91–1.1) | 0.99 (0.90–1.0) | 1.0 (0.94–1.1) | 0.24 |

In the Cox proportional hazards regression models, the median of each quartile was used as a continuous variable to assess the overall trends in HRs beyond the quartiles of spice intake.

[a]: The crude model was adjusted for age, sex.

[b]: The model was adjusted for age, sex, BMI, smoking, education level, physical activity, total energy, saturated fat (percentage of energy), and fiber intake (gr/1000 kcal).

[c]: In the models used to estimate the HR for high BP, high, high FBG, and low HDL-C, the continuous values of each risk factor at the beginning of the study were included as adjustment factors.

[d]: Model adjusted for age, sex, diabetes risk score, physical activity, smoking, education level, total energy intake, saturated fat (percentage of energy), and fiber intake (gr/1000 kcal).

HR: hazard ratio, FBG: fasting blood glucose, HDL-C: high-density lipoprotein, TG: triglyceride, WC: waist circumference, BP: blood pressure, T2D: type 2 diabetes.

not alter after adjustment for confounding variables [HR: 1, 1.0 (0.94–1.1), 1.0 (0.91–1.0), 0.98 (0.89–1.0), P-trend = 0.8].

In the analysis for MetS incidence, the baseline features and dietary intake of subjects among quartiles of spice are demonstrated in S1 and S2 Tables, respectively. The findings showed that subjects in the upper quartile of spice intake had lower TG, FBG, WC, and diastolic BP than the lower quartiles. Also, HDL-C was higher in the higher quartile of spice intake than in the lower quartiles. Total energy intake, total fat, MUFA, PUFA, and SFA intake

**Table 4.** HRs (95% CI) of T2D, MetS, and its risk factors incidence beyond categories of pepper intake in adult participants of TLGS.

| | Variables | Q1 | Q2 | Q3 | Q4 | P-trend |
|---|---|---|---|---|---|---|
| | | | Quartiles of Pepper intake* | | | |
| **Incidence of MetS (n = 4353)** | | 260 | 324 | 329 | 298 | |
| **Median** (gr/day) (Min-Max) | | 0.88(0.0–3.0) | 3.2 (3–6.1) | 8.2 (6.1–13.1) | 19.7 (13.0–184) | |
| HR (95% CI) | [a] Crude | Ref. | 1.06 (0.97–1.15) | 0.99 (0.91–1.08) | 1.07 (0.98–1.17) | 0.34 |
| | [b] Model adjusted | Ref. | 0.98 (0.88–1.09) | 1.00 (0.90–1.11) | 1.02 (0.91–1.13) | 0.51 |
| **Incidence of Low HDL-C (n = 2688)** | | 234 | 237 | 236 | 260 | |
| **Median** (gr/day) | | 0.90 | 3.19 | 8.23 | 19.9 | |
| HR (95% CI) | [a] Crude | Ref | 0.97 (0.87–1.09) | 0.87 (0.78–0.97) | 1.03 (0.92–1.14) | 0.008 |
| | [c] Model adjusted | Ref | 0.91 (0.79–1.05) | 0.89 (0.78–1.01) | 1.01 (0.88–1.16) | 0.24 |
| **Incidence of High FBG(n = 4949)** | | 305 | 355 | 342 | 305 | |
| **Median** (gr/day) | | 0.90 | 3.19 | 8.33 | 19.7 | |
| HR (95% CI) | [a] Crude | Ref | 1.01 (0.93–1.09) | 0.93 (0.86–1.01) | 0.98 (0.90–1.06) | 0.88 |
| | [c] Model adjusted | Ref | 0.98 (0.88–1.08) | 0.94 (0.85–1.04) | 0.96 (0.87–1.07) | 0.54 |
| **Incidence of High TG (n = 4011)** | | 300 | 326 | 343 | 337 | |
| **Median** (gr/day) | | 0.81 | 3.19 | 8.32 | 19.8 | |
| HR (95% CI) | [a] Crude | Ref | 1.00 (0.92–1.09) | 0.98 (0.90–1.07) | 1.03 (0.94–1.13) | 0.50 |
| | [c] Model adjusted | Ref | 0.94 (0.84–1.05) | 0.99 (0.88–1.10) | 0.99 (0.88–1.11) | 0.31 |
| **Incidence of High WC (n = 3709)** | | 231 | 243 | 247 | 260 | |
| **Median** | | 0.89 | 3.19 | 8.21 | 19.7 | |
| HR (95% CI) | [1] Crude | Ref | 0.99 (0.91–1.09) | 0.95 (0.87–1.04) | 1.04 (0.95–1.14) | 0.29 |
| | [c] Model adjusted | Ref | 0.92 (0.82–1.04) | 0.92 (0.82–1.03) | 0.94 (0.83–1.06) | 0.21 |
| **Incidence High BP (n = 4754)** | | 291 | 351 | 352 | 301 | |
| **Median** (gr/day) | | 0.89 | 3.19 | 8.22 | 19.7 | |
| HR (95% CI) | [a] Crude | Ref | 1.04 (0.96–1.13) | 1.00 (0.92–1.09) | 1.02 (0.94–1.11) | 0.57 |
| | [c] Model adjusted | Ref | 0.91 (0.82–1.01) | 0.97 (0.88–1.07) | 0.89 (0.80–0.99) | 0.20 |
| **Incidence T2D (n)** | | 107 | 95 | 103 | 101 | |
| **Median** (gr/day) (Min-Max) | | 1.4 (0.0–3.0) | 4.0 (3.0–6.4) | 8.6 (6.4–13.1) | 19.7 (13.1–275.4) | |
| HR (95% CI) | [a] Crude | Ref | 0.97 (0.90–1.0) | 0.96 (0.89–1.0) | 0.95(0.88–1.0) | 0.80 |
| | [d] Model adjusted | Ref | 0.98 (0.89–1.0) | 1.0 (0.91–1.0) | 0.98 (0.89–1.0) | 0.86 |

In the Cox proportional hazards regression models, the median of each quartile was used as a continuous variable to assess the overall trends in HRs beyond the quartiles of pepper intake.

[a]: The crude model was adjusted for age, sex.

[b]: The model was adjusted for age, sex, BMI, smoking, education level, physical activity, total energy, saturated fat (percentage of energy), and fiber intake (gr/1000 kcal).

[c]: In the models used to estimate the HR for high BP, high, high FBG, and low HDL-C, the continuous values of each risk factor at the beginning of the study were included as adjustment factors.

[d]: Model adjusted for age, sex, diabetes risk score, physical activity, smoking, education level, total energy intake, saturated fat (percentage of energy), and fiber intake (gr/1000 kcal).

HR: hazard ratio, FBG: fasting blood glucose, HDL-C: high-density lipoprotein, TG: triglyceride, WC: waist circumference, BP: blood pressure, T2D: type 2 diabetes.

*Pepper = Bell pepper + green pepper.

increased in the upper quartiles of spice. Moreover, protein and carbohydrate intakes decreased in the upper quartiles of spice consumption.

Spice intake had no association with MetS incidence in the crude and adjusted [HR: 1.01 (0.91–1.12), 1.02 (0.92–1.14), 1.05 (0.94–1.18) P trend = 0.65] models (Table 3). Pepper consumption had no association with MetS incidence in the crude and adjusted model (Table 4). Moreover, the results showed that by increasing spice intake, the HR of BP incidence increased

in the crude and adjusted [HR: 1, 1.03 (0.93–1.14), 1.06 (0.96–1.17), 1.16 (1.04–1.30), P-trend = 0.007] models. Also, by increasing pepper intake, the HR of low HDL-C level significantly decreased in quartile 3 of the crude model [HR: 1, 0.97 (0.87–1.09), 0.87 (0.78–0.97), 1.03 (0.92–1.14), P-trend = 0.008].

## Discussion

This study evaluated the association of spice and pepper consumption with T2D, MetS, and related components incidence. The findings showed that spice and pepper intakes were not associated with T2D and MetS. However, spice intake significantly increased T2D incidence in the crude model. Also, by increasing spice intake, the HRs of high BP and high TG increased. In addition, pepper intake significantly improved low HDL-C in the crude model.

Until now, no prospective study has yet examined the relation between spice and pepper intakes and the incidence of diabetes and MetS.

Although we did not find a study to show the mean intake of spice consumption in Iran, a cohort study in Iran showed that the mean consumption of curcumin is 3.4 mg/day in the Golestan population [31]. While the clinical trials use pharmaceutical doses of spice, the usual intake is less than spice's beneficial effects, as shown in RCTs [32]. A review showed that cinnamon can improve insulin resistance in more than 3 gr per day [33]. Even among the Indian population, known for its preference for spicy foods, the median intake of curcumin ranks from 0.004 to 0.1 g per day, while the intake of red pepper is between 2.4 to 4.1 g per day for adults. Consequently, the health advantages associated with spice consumption may not be attainable with the typical amounts used in home cooking [34].

An RCT was conducted to evaluate the effect of Turmeric (2.4 g/day) and Black seed (1.5 g/day) in the mixture (900 mg Black seeds and 1.5 g Turmeric/day) or alone for eight weeks in patients with MetS. The results showed that combination therapy can improve CRP, LDL-C, HDL-C, FBG, and total cholesterol, in comparison to the placebo group. Turmeric also significantly decreased CRP, LDL-C, and total cholesterol. Black seed reduced TG and total cholesterol significantly [35]. Our data revealed that pepper or spice consumption was not associated with MetS incidence; however, our findings showed that pepper intake can improve the HR of low HDL-C levels. Also, our study exhibited that in the upper quartile of spice intake, the HR of high TG level increased remarkably. The findings showed that by increasing spice intake, fat consumption has increased. Although we have adjusted to control for fat intake, it may not control the effects of excessive fat consumption, increasing triglycerides. These findings suggest that the specific amount and type of spice or pepper can have different effects.

Oxidative stress and inflammatory conditions are related to the MetS. Cinnamon (*Cinnamomum*) regulates gene expression of pro-inflammatory and anti-inflammatory and glucose transporters in mouse macrophages and adipocytes. Also, curcumin derived from turmeric (*Curcuma longa*) increased glucose uptake through insulin sensitivity improvement in 3T3-L1 cells. Further, curcumin has beneficial anti-inflammatory effects and glucose regulatory properties by suppressing Tumor necrosis factor (TNF) and IL-6 transcription and secretion [36].

A prospective cohort study indicated that both the frequency and average consumption of spicy foods were conversely related to the LDL-C to cholesterol ratio. Additionally, this study proposes that spicy foods may be a substitute strategy to alleviate the risk of CVD. This indicates that the bioactive components in spicy food can justify these metabolic effects. However, the frequency and usual amount of spicy foods are positively related to triacylglycerol [37]. In contrast, another clinical study demonstrated that the chili diet has no remarkable effects on lipid metabolism after three weeks of intervention [38]. Glucose uptake is facilitated by

AMPK, PI3K or MAPK signaling pathways. Also, curcumin activates peroxisome proliferator-activated receptor (PPAR), suppressing LDL-C receptor gene expression [39].

A meta-analysis exhibited that ginger (tablet, capsules, powder, or rhizomes) drastically decreased FBG and TG and improved HDL-C levels [40]. Another meta-analysis demonstrated that cinnamon significantly dwindled LDL-C, cholesterol, TG, and elevated HDL-C levels, although HbA1c had no remarkable change [41]. Cellular cholesterol reduction decreases fatty acid synthesis by inhibiting fatty acid synthase. Cholesterol reduction defeats adipocyte differentiation and lipid accumulation, attributed to curcumin benefits [33]. Herbs and spices may have beneficial metabolic effects on AMPK activating, including fatty acid oxidation and mitochondrial functions which help to attenuate MetS symptoms [34].

In a similar study, the intake of polyphenols and flavonoids from fruits and vegetables with the incidence of gestational diabetes mellitus (GDM) has been evaluated. The results showed that the total dietary intake of polyphenols and flavonoids from fruits decreased the risk of GDM. In contrast, the polyphenols and flavonoids in vegetables did not relate to GDM. This reduction is mainly associated with 2-h FBG rather than FBG and 1-h FBG [42]. Nonetheless, our findings did not show any relation between spice consumption and diabetes incidence. Spices have polyphenols that can affect glucose metabolism, including intestinal glucose absorption, insulin liberation from B-cells in the pancreas, regulation of glucose liberation from the liver, increased expression of insulin receptors, and enhanced glucose uptake by insulin-sensitive tissues [43]. However, these effects can be altered during digestion and absorption. Furthermore, most polyphenols contained in spices have no sufficient absorption, and plasma levels have fluctuated owing to their quick metabolism. The complexity effects of spices on T2D have been determined by the fact that different spices contain different phenolic compounds and synergistic effects [44].

LV et al. study [45] reported that spicy food consumption strongly reverses specific causes of mortality independent of other risk factors of death in women and men. Also, this study illustrated that those who consumed spicy foods six or seven days per week experienced a 14% lessen in the total mortality rate. However, this relation was found to be stronger among those who did not consume alcohol compared to those who did.

As a result, the effects of spices on diabetes can vary significantly based on the types and combinations of spices used. Hence, it is crucial to consider the specific types and mixtures of spices when evaluating their impact on diabetes [46]. Thus, the kind of spice utilized, the timing of its addition to the food, and the cooking methods employed can influence the polyphenols and their antioxidant properties. These factors may lead to diverse outcomes in research studies.

Similar to a previous study, a prospective study investigated the relationship between total polyphenol intake and its subtypes with diabetes incidence. Results indicated that the intake of polyphenols, phenolic acids, and lignins had an inverse relation with T2D. This investigation also showed that several polyphenols can decrease the incidence of diabetes [47]. As a result of two previous studies, such polyphenols (not all of them) can reduce diabetes incidence.

Hashemian et al. [31] accomplished a cohort study to assess the association between some spice intake (turmeric, black or chili pepper, cinnamon, and saffron) with all and specific causes of mortality in Iran. The findings showed that turmeric and saffron consumption drastically reduced overall mortality and cardiovascular mortality. Furthermore, black or chili pepper was associated with a remarkable decrease in overall mortality risk. Also, no spices have shown a relationship with cancer mortality. In our study, we evaluated the spices that people commonly consume. However, Hashemian's research assessed the influence of

specific spices on certain aspects of mortality rates. Also, our results showed that BP significantly increased in the upper quartiles of spices. Also, individuals who consumed more spices tended to have a higher fat intake. Based on our results, this can justify that other components of foods can change the benefit of spices and may change polyphenol and antioxidant effects, leading to undesirable results. A meta-analysis represented that the intake of a remarkable amount of phenolic acid (600 mg per day) is related to a 34% reduction in the risk of T2D. Thus, we conclude that the dosage of polyphenol consumption is essential for its effectiveness [48]. In our study, the inadequate spice intake may have led to unremarkable results. In an RCT, consuming cinnamon with a dosage of 1, 3, and 6 grams per day can remarkably diminish serum levels of Glucose, TG, and LDL-C compared to placebo in people with T2D [49]. Another RCT assessed the effect of ginger consumption (3 grams/day) on glycemic indices. The result of this study indicated that ginger can reduce FBG and HBA1c and elevate the QUICKI index in the intervention group in comparison to the placebo group [50].

Also, Chuengsamarn et al. [51] conducted an RCT to measure the effect of curcumin intake on pre-diabetic patients to prevent diabetic conditions. In this 9-month intervention, the curcumin group exhibits better pancreatic B-cell function. Also, C-peptide and HOMA-IR were lower in the treatment group, and adiponectin and HOMA-B were higher in the curcumin group. As a result, the curcumin intervention lowered the development of pre-diabetic to diabetes conditions.

In an in vitro study, the effect of some high spice consumption on hemoglobin glycation has been assessed. The results suggest that wild caraway, turmeric, cardamom, and black pepper may prevent hemoglobin glycation. However, some agents such as anise and saffron have preventive and pro-glycation properties. Based on the results obtained, wild caraway, turmeric, cardamom, and black pepper, particularly wild caraway extracts, are powerful antiglycation agents, that can contribute to preventing diabetic complications associated with glycation [52]. Consequently, results showed that various factors can affect the bioavailability and absorption of the polyphenol or antioxidant activity of spices. So, further studies are needed to approve these matters.

Our result exhibited several notable strengths. First, the TLGS provided robust data on the urban population. Second, the diagnosis of T2D was based on an FBG test, ensuring greater reliability. Third, the nutrient content and food items were assessed cumulatively from the baseline survey through all follow-up visits until outcomes were diagnosed. Lastly, we conducted a comprehensive evaluation of all dietary factors, including fiber and fatty acids, in relation to the onset of outcomes, while accounting for potential confounding factors in our final analysis.

We must acknowledge various limitations in our study. Firstly, we needed more data on the common methods of using spices and the time of adding them to food, and thus, we failed to analyze their impact on the development of T2D and Mets. Secondly, despite our efforts to identify all confounding, this may not have been eliminated due to knowledge or measurement gaps. Furthermore, since insulin sensitivity- a highly sensitive marker was unavailable for all participants, we could not detect the association between spice and pepper intakes and the risk of insulin resistance. Also, the method of assessing spice intake was not validated or tested for reliability/repeatability. Dietary information was collected by expert dietitians without the use of food replicas. Previous studies utilizing (FFQ) have shown significant under-reporting of energy intake at the group level. The estimates of under-reporting range from 4.6% to 42% when compared to the doubly labeled water method [53]. The level of misreporting observed in our study was in this rank.

## Conclusion

Finally, our results showed that spice and pepper consumption had no relation with the incidence of T2D and MetS. In addition, the HR of low HDL-C improved in the third quartile of pepper. Also, increasing spice intake significantly increased the HRs of BP and TG levels.

## Supporting information

**S1 Table. Baseline characteristics of adult participants across quartiles of spice intake: The Tehran Lipid and Glucose Study (Mets).**
(DOCX)

**S2 Table. Dietary intake across quartiles of spice- intake (Mets).**
(DOCX)

## Acknowledgments

The authors would like to represent their gratitude to the participants and the personnel of the TLGS for their collaboration.

## Author Contributions

**Conceptualization:** Fereidoun Azizi.

**Software:** Azam Ildarabadi, Shahrzad Daei.

**Writing – original draft:** Azam Ildarabadi, Firoozeh Hosseini-Esfahani.

**Writing – review & editing:** Firoozeh Hosseini-Esfahani, Parvin Mirmiran, Fereidoun Azizi.

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
