## [Decision Letter · Decision Letter 0]

22 Jul 2024

PONE-D-24-24401Associations between Spice or Pepper (Capsicum annuum) Consumption and Diabetes or Metabolic Syndrome IncidencePLOS ONE

Dear Dr. Hosseini-Esfahani,

Thank you for submitting your manuscript to PLOS ONE. After careful consideration, we feel that it has merit but does not fully meet PLOS ONE’s publication criteria as it currently stands. Therefore, we invite you to submit a revised version of the manuscript that addresses the points raised during the review process.

 Especially, 1) please address reviewers' comments. 2) please correct the grammar errors. Please ensure that your decision is justified on PLOS ONE’s publication criteria and not, for example, on novelty or perceived impact.

We look forward to receiving your revised manuscript.

Kind regards,

Guoying Wang, MD, PhD

Academic Editor

PLOS ONE

Journal Requirements:

2. Thank you for stating the following financial disclosure: "This work was supported by the Research Institute for Endocrine Science, Shahid Beheshti University of Medical Science (Tehran, Iran) under Grant number. 43003670" 

3. In the online submission form, you indicated that the datasets generated and analyzed during the current study are not publicly

available because data contains sensitive individual information and data are owned

by Research Institute for Endocrine Sciences, Shahid Beheshti University of Medical Sciences. Tehran, Iran

Data are available from the ethics committee of the Research Institute for Endocrine Sciences, Shahid Beheshti

University of Medical Sciences, Tehran, Iran, whenever data request has

been sent.

No.24, Arabi Street, Yemen Avenue, Chamran Highway

Fax: +98 (21)22402463

Postal code: 1985717413

Email: info@endocrine.ac.ir.

Reviewers' comments:

Reviewer's Responses to Questions

**Comments to the Author**

1. Is the manuscript technically sound, and do the data support the conclusions?

Reviewer #1: Yes

Reviewer #2: Yes

2. Has the statistical analysis been performed appropriately and rigorously? 

Reviewer #1: Yes

Reviewer #2: Yes

3. Have the authors made all data underlying the findings in their manuscript fully available?

Reviewer #1: No

Reviewer #2: No

4. Is the manuscript presented in an intelligible fashion and written in standard English?

Reviewer #1: No

Reviewer #2: Yes

5. Review Comments to the Author

Reviewer #1: The authors report on the relationship between spice and pepper consumption and metabolic syndrome. The use of a cohort study to gather the data is appropriate and the participant flow is clearly depicted. The statistical analysis is appropriate and multiple variables are included. Research on the health effects of spice intake is limited, so this research can potentially contribute to the evidence base.

It is unclear why the authors chose to evaluate the health effects of dietary Capsicum annuum (as opposed to other food sources of polyphenols) and spices. Is pepper a heavily consumed food in Iran? Often, spice and herb intakes are assessed together.

My main concern is the lack of a clear description of the spice and pepper intake assessment. Two papers are cited describing validation of a TLGS FFQ for food group and nutrient analysis. There is no description of the questions about spice and pepper intake - specific spices, portion sizes, and frequency of intake. Apparently the assessment method of spices and pepper (Capsicum annuum) intake has not been validated. A table showing the spices assessed and intakes of each is needed. Table 4 should show the quartile levels of pepper intake. Were intakes of Capsicum annuum and bell/green pepper combined?

English editing is required. There are multiple errors in sentence structure, punctuation and the use of articles (a, the).

Reviewer #2: The manuscript is written well and can be published with a few amendments. The research is meaningful for field of T2D and the use of spices in the management of NCDs. There some language errors but the English is acceptable. Sugggested changes are in the Reviewer attachments.

6. PLOS authors have the option to publish the peer review history of their article (what does this mean?). If published, this will include your full peer review and any attached files.

Reviewer #1: No

Reviewer #2: **Yes: **Tertia Van Zyl

---

## [Author Response · Author response to Decision Letter 0]

4 Aug 2024

Dear Dr. Guoying Wang

I would like to express my gratitude for your valuable time and thoughtful comments on our manuscript titled “Associations between Spice or Pepper (Capsicum annuum) Consumption and Diabetes or Metabolic Syndrome Incidence.” Your insights have been instrumental in refining our work.

In response to your suggestions, we have diligently revised the manuscript. To facilitate your review, we have provided a point-by-point summary of the requested changes, and the modifications are highlighted using the track changes feature in the submitted file.

We are hopeful that the revised manuscript now aligns with the standards for publication in the " PLOS ONE " Your interest in our research and the opportunity to enhance its quality are greatly appreciated. We remain open to any additional feedback or further revisions that you may recommend ensuring the manuscript meets the journal's criteria.

Once again, thank you for your commitment to advancing scientific discourse, and we look forward to your feedback on the revised version.

Please find the below comments/questions by the editor/reviewers (in black color) and our response/clarification/explanation to the comments and questions (in red color).

Best regards,

Parvin Mirmiran, Prof.

Nutrition and Endocrine Research Center, Research Institute for Endocrine Sciences, Shahid Beheshti University of Medical Sciences, Tehran, Iran

E-mail: parvin.mirmiran@sbmu.ac.ir

Parvin.mirmiran@gmail.com

Journal Requirements:

1. Thank you for stating the following financial disclosure: "This work was supported by the Research Institute for Endocrine Science, Shahid Beheshti University of Medical Science (Tehran, Iran) under Grant number. 43003670" 

Reply: Agree and corrected. This has been added to funding section. Page 28, line 467-468.

2. In the online submission form, you indicated that the datasets generated and analyzed during the current study are not publicly available because data contains sensitive individual information and data are owned by Research Institute for Endocrine Sciences, Shahid Beheshti University of Medical Sciences. Tehran, Iran. Data are available from the ethics committee of the Research Institute for Endocrine Sciences, Shahid Beheshti

University of Medical Sciences, Tehran, Iran, whenever data request has

been sent.

No.24, Arabi Street, Yemen Avenue, Chamran Highway

Fax: +98 (21)22402463 Postal code: 1985717413

Email: info@endocrine.ac.ir.

The datasets generated and analyzed during the current study are not publicly available for ethical or legal reasons due to confidentiality agreements and data are owned by Research Institute for Endocrine Sciences, Shahid Beheshti University of Medical Sciences. Tehran, Iran. Data are available from the ethics committee of the Research Institute for Endocrine Sciences, Shahid Beheshti University of Medical Sciences, Tehran, Iran, whenever data request has been sent.

No.24, Arabi Street, Yemen Avenue, Chamran Highway

Fax: +98 (21)22402463 Postal code: 1985717413

Email: info@endocrine.ac.ir.

Review Comments to the Author

1. Reviewer #1: The authors report on the relationship between spice and pepper consumption and metabolic syndrome. The use of a cohort study to gather the data is appropriate and the participant flow is clearly depicted. The statistical analysis is appropriate and multiple variables are included. Research on the health effects of spice intake is limited, so this research can potentially contribute to the evidence base.

It is unclear why the authors chose to evaluate the health effects of dietary Capsicum annuum (as opposed to other food sources of polyphenols) and spices. Is pepper a heavily consumed food in Iran? Often, spice and herb intakes are assessed together.

My main concern is the lack of a clear description of the spice and pepper intake assessment. Two papers are cited describing validation of a TLGS FFQ for food group and nutrient analysis. There is no description of the questions about spice and pepper intake - specific spices, portion sizes, and frequency of intake. Apparently the assessment method of spices and pepper (Capsicum annuum) intake has not been validated. A table showing the spices assessed and intakes of each is needed. Table 4 should show the quartile levels of pepper intake. Were intakes of Capsicum annuum and bell/green pepper combined?

Reply:

Metabolic syndrome is a main factor that contributes to various chronic diseases such as diabetes, as a public health disorder. Based on lectures, polyphenols and antioxidants can improve health status, therefore the aim of this study was to evaluate the ingredients that have polyphenols content in the usual Iranian foods. We decided to assess spices and pepper as a main polyphenol that we added to prepare our foods. This study evaluated usual spices intake in Iran including turmeric, cinnamon, pepper, saffron, and ginger. Also, peppers, green, and bell pepper combined, were reported to grams/day. Spices and pepper intake (green pepper and bell pepper) were analyzed separately in statistical analysis. 

This section has been added concisely in the introduction section. Page 5, line 80-82

The mean intake of spice is 1.6±1.4 grams per day. Also, the mean intake of pepper is 10.1±13.5 grams per day. Page 12, line 234-235.

Although we did not find a study to show the mean intake of spice consumption in Iran, a cohort study in Iran showed that the mean intake of curcumin is 3.4 mg/day in the Golestan population [32]. While the clinical trials use pharmaceutical doses of spice, the usual intake is less than the beneficial effects of spice, which was shown in RCTs [33]. A review showed that cinnamon can improve insulin resistance in more than 3 gr per day [34]. Even in the Indian population, where spicy foods are popular, the mean intake of curcumin is 0.004 to 0.1 g/d, and red pepper is 2.4 to 4.1 g/d for adults. Therefore, the health benefits of spice consumption may not be achievable with the usual intake used for domestic purposes [35]. we discuss the spice intake in the discussion section. Page 22, line 321-328..

In the FFQ questionnaire, usual spice intake was questioned as a food item. The interviewer asked to report the usual spice intake as teaspoons/day and transformed to gram/day. For green and bell peppers, the interviewer asked to report how many they use, and then they transformed intakes to grams in a day. Page 7, line 126-129.

In checking the validity and reliability of the questionnaire, peppers were examined with other vegetables as a group; however, the validity and reliability of spice intake were not assessed. Page 7, line 122-124.

English editing is required. There are multiple errors in sentence structure, punctuation and the use of articles (a, the).

Reply:

Agreed and corrected.

Reviewer #2: The manuscript is written well and can be published with a few amendments. The research is meaningful for field of T2D and the use of spices in the management of NCDs. There some language errors but the English is acceptable. Suggested changes are in the Reviewer attachments.

Review PONE-D-24-24401

1. Line 50: Suggest to rather use …and many more than and etc.

Reply: Thank you for your comments. It has been Corrected. Page 4, line 50) 

2. Line 51: How does MetS manifests with WC – is there a certain reference value? Or will abdominal obesity be more specific?

Reply: This has been changed to high WC. Page 4, line 51.

3. Line 61-61: Improve the sentence highlighted in yellow Further, the using of herbs and spices has been paid more attention, while the modern medicine has been improved.

Reply: Agreed and corrected. Page 4, lines 61-62. 

Although modern medicine has made great progress, using herbs and spices as complementary medicine has attracted much attention.

4. Line 92: Add the word highlighted in red. Of individuals participating in survey 4, 6882 subjects aged ≥18 years were? randomly selected for dietary assessment.

Reply: Agreed and corrected. Page 6, line 94. 

5. Line 101-102: Add the words highlighted in red. Subjects participating in survey 4 (baseline), aged ≥18 years, and who completed the dietary assessment were entered in this study for MetS secondary analysis.

Reply: Agreed and corrected. Page 6, lines 104-105. 

6. Line 106: Add the word highlighted in red. …the baseline population and they were tracked until survey 6.

Reply: Agreed and corrected. Page 6, line 110. 

7. Line 102-104: Subjects with MetS diagnosis at baseline [20], pregnant or lactating women were excluded. Also, individuals with over- or under reporting of energy intake (≥4,200 or <800 kcal/day) were omitted [19] – how is this different form line 93-97? Or is this a repeat?

Reply: These are two lines for selecting individuals for the association between spice or pepper intakes and the incidence of MetS and diabetes. Page 6, lines 101-102 and 105-106. 

8. Line 200-201: Is the fat included there total fat? 

Reply: Yes, agreed and corrected. Page 11, line 209.

9. Table 1: What are the cut-offs for the different quartiles? What was the average spice intake per quartile? Does the spice intake include the pepper intake as well? Please amend the table title to include peppers. Should you merge the cells for the p-value for education?

Reply: In Table 1, we showed the relation of baseline characteristics with T2D incidence in each quartile of spice only, without pepper intake. The baseline characteristics were reported for spice intake only.

Cut off: The maximum and minimum amounts of pepper and spice have been added to Table 3 and Table 4.

The Mean±SD has been added to Tables 1 and Table 2 which showed the average spice intake per quartile.

Also, the cell for p-value has been merged. 

10. Table 2&3 footnote: HR explanation/definition

Reply: Agree and corrected., this has been added to the footnote. Page 18 and 21, lines 282 and 310.

11. Table 4: Should it be Quartiles of Pepper intake*

Reply: It has been added to Table 4. 

12. Line 309: replace didn’t with did not

Reply: Agreed and corrected. Page 22, line 315.

13. Line 324-326: Add the words highlighted in red. Although, we have adjusted to control for the intake of fat, it is possible that we have not been able to control for the effects of excessive fat consumption, resulting in an increase in triglycerides. 

Reply: Agreed and corrected. Page 23, line 338-339. 

14. Line 354: replace didn’t with did not

Reply: Agreed and corrected. Page 24, line 368. 

15. Line 355: replace didn’t with did not

Reply: Agreed and corrected. Page 24, line 370.

16. Line 360: …these effects can be altered during the process of digestion and absorption

Reply: Agreed and corrected. Page 24, line 374.

17. Line 360: The complexity effects of spices on diabetes T2D has

Reply: Agreed and corrected. Page 24, line 377.

18. Line 367: 367 illustrated that those who consumed spicy food 6 six or 7 seven days a week

Reply: Agreed and corrected. Page 24, line 381. 

19. Line 368: replace didn’t with did not

Reply: Agreed and corrected. Page 25, line 382.

20. Line 396: Perhaps, in our study, the inadequate amount of spice intake leading led to unremarkable results.

Answer: The word has been changed. Page 26, line 410.

21. Line 401: HBA1C replace with HbA1c, check the rest of the manuscript for correct format.

Answer: It has been replaced. Page 26, line 414.

22. Line 429: Add the word highlighted in red. Secondly, despite our efforts to identify all cofounding this may not have been eliminated due to knowledge or measurement gaps.

Answer: The highlighted word has been added. Page 27, line 442.

23. Line 439: …of T2D and Mets MetS. 

Answer: It has been changed. Page 28, line 452.

24. FBS vs FBG can you use it interchangeable. The first part of the article it is FBG and at the results it becomes FBS.

Answer: In the article, all FBS have been changed to FBG.

---

## [Decision Letter · Decision Letter 1]

3 Sep 2024

PONE-D-24-24401R1Associations between Spice or Pepper (Capsicum annuum) Consumption and Diabetes or Metabolic Syndrome IncidencePLOS ONE

Dear Dr. Hosseini-Esfahani,

Thank you for submitting your manuscript to PLOS ONE. After careful consideration, we feel that it has merit but does not fully meet PLOS ONE’s publication criteria as it currently stands. Therefore, we invite you to submit a revised version of the manuscript that addresses the points raised during the review process.

Please address reviewers' comments.

We look forward to receiving your revised manuscript.

Kind regards,

Guoying Wang, MD, PhD

Academic Editor

PLOS ONE

Journal Requirements:

Reviewers' comments:

Reviewer's Responses to Questions

**Comments to the Author**

1. If the authors have adequately addressed your comments raised in a previous round of review and you feel that this manuscript is now acceptable for publication, you may indicate that here to bypass the “Comments to the Author” section, enter your conflict of interest statement in the “Confidential to Editor” section, and submit your "Accept" recommendation.

Reviewer #1: All comments have been addressed

Reviewer #2: All comments have been addressed

2. Is the manuscript technically sound, and do the data support the conclusions?

Reviewer #1: Yes

Reviewer #2: Yes

3. Has the statistical analysis been performed appropriately and rigorously? 

Reviewer #1: Yes

Reviewer #2: Yes

4. Have the authors made all data underlying the findings in their manuscript fully available?

Reviewer #1: No

Reviewer #2: No

5. Is the manuscript presented in an intelligible fashion and written in standard English?

Reviewer #1: Yes

Reviewer #2: Yes

6. Review Comments to the Author

Reviewer #1: The authors have addressed most of my concerns. One additional revision is to add to the limitations in the discussion that the method of assessing spice intake was not validated or tested for reliability/repeatability.

Reviewer #2: Dear Author,

Thank you for the improvements in the manuscript. Here are just some minor suggestions:

Line 130-131: Portion sizes were then converted to mass (gr/day) from the household measurements. Would this be a better statement?

Line 133: National foods not listed in the USDA FCT were provided from the Iranian FCT. Would this be a better statement?

Line 228: This sentence should probably read: Furthermore, increased spice consumption was associated with lower systolic and diastolic BP (P<0.001). The study design is not able to support causality.

Line 360 and 414: HbA1c not corrected

7. PLOS authors have the option to publish the peer review history of their article (what does this mean?). If published, this will include your full peer review and any attached files.

Reviewer #1: No

Reviewer #2: **Yes: **Tertia van Zyl

---

## [Author Response · Author response to Decision Letter 1]

12 Sep 2024

Dear Dr. Guoying Wang

I would like to express my gratitude for your valuable time and thoughtful comments on our manuscript titled “Associations between Spice or Pepper (Capsicum annuum) Consumption and Diabetes or Metabolic Syndrome Incidence.” Your insights have been instrumental in refining our work.

In response to your suggestions, we have diligently revised the manuscript. To facilitate your review, we have provided a point-by-point summary of the requested changes, and the modifications are highlighted using the track changes feature in the submitted file.

We are hopeful that the revised manuscript now aligns with the standards for publication in the "PLOS One." Your interest in our research and the opportunity to enhance its quality are greatly appreciated. We remain open to any additional feedback or further revisions that you may recommend ensuring the manuscript meets the journal's criteria.

Once again, thank you for your commitment to advancing scientific discourse, and we look forward to your feedback on the revised version.

Please find the below comments/questions by the editor/reviewers (in black color) and our response/clarification/explanation to the comments and questions (in blue color).

Best regards,

Parvin Mirmiran, Prof.

Nutrition and Endocrine Research Center, Research Institute for Endocrine Sciences, Shahid Beheshti University of Medical Sciences, Tehran, Iran

E-mail: parvin.mirmiran@sbmu.ac.ir

Parvin.mirmiran@gmail.com

Journal Requirements:

Reply: It has been checked. 

Reviewer #1:

 The authors have addressed most of my concerns. One additional revision is to add to the limitations in the discussion that the method of assessing spice intake was not validated or tested for reliability/repeatability.

Reply|: Based one your request, it has been added to discussion. Also, the method of assessing spice intake was not validated or tested for reliability/repeatability. Page 27, line 446-447.

Reviewer #2:

Line 130-131: Portion sizes were then converted to mass (gr/day) from the household measurements. Would this be a better statement?

Reply: This sentence has been changed based your comment. Page 7, line 130-131.

Line 133: National foods not listed in the USDA FCT were provided from the Iranian FCT. Would this be a better statement?

Reply: Based on your suggestion, this sentence has been changed. Page 7, line 133.

Line 228: This sentence should probably read: Furthermore, increased spice consumption was associated with lower systolic and diastolic BP (P<0.001). The study design is not able to support causality.

Reply: It has been changed. Page 11-12, line 228-229.

Line 360 and 414: HbA1c not corrected.

Reply: It has been corrected. Page 24, line 361.

---

## [Decision Letter · Decision Letter 2]

30 Oct 2024

PONE-D-24-24401R2Associations between Spice or Pepper (Capsicum annuum) Consumption and Diabetes or Metabolic Syndrome IncidencePLOS ONE

Dear Dr. Hosseini-Esfahani,

Thank you for submitting your manuscript to PLOS ONE. After careful consideration, we feel that it has merit but does not fully meet PLOS ONE’s publication criteria as it currently stands. Therefore, we invite you to submit a revised version of the manuscript that addresses the points raised during the review process.

**Please review and address the new reviewer’s comments. The primary concern preventing from acceptance of your manuscript is that the questionnaire used for assessing spice intake has not been validated. Additionally, after reviewing the 168-item FFQ, I noticed that it does not include "spices" in powder form. The only items related to spices are Jalapeño and capsicum. Therefore, please provide the 168-item questionnaire you used along with your revised manuscript (translation into English is not necessary). Also, ensure that the similarity score of your manuscript is reduced to below 20% before resubmission.**

We look forward to receiving your revised manuscript.

Kind regards,

Zohreh Sajadi Hezaveh

Academic Editor

PLOS ONE

**Journal Requirements:**

Reviewers' comments:

Reviewer's Responses to Questions

**Comments to the Author**

1. If the authors have adequately addressed your comments raised in a previous round of review and you feel that this manuscript is now acceptable for publication, you may indicate that here to bypass the “Comments to the Author” section, enter your conflict of interest statement in the “Confidential to Editor” section, and submit your "Accept" recommendation.

Reviewer #1: All comments have been addressed

2. Is the manuscript technically sound, and do the data support the conclusions?

Reviewer #1: Yes

3. Has the statistical analysis been performed appropriately and rigorously? 

Reviewer #1: Yes

4. Have the authors made all data underlying the findings in their manuscript fully available?

Reviewer #1: Yes

5. Is the manuscript presented in an intelligible fashion and written in standard English?

Reviewer #1: Yes

6. Review Comments to the Author

**Reviewer #1: **The authors addressed my comment about stating the limitation of using a questionnaire that has not been validated.

7. PLOS authors have the option to publish the peer review history of their article (what does this mean?). If published, this will include your full peer review and any attached files.

Reviewer #1: No

---

## [Author Response · Author response to Decision Letter 2]

10 Nov 2024

I would like to express my gratitude for your valuable time and thoughtful comments on our manuscript titled “Associations between Spice or Pepper (Capsicum annuum) Consumption and Diabetes or Metabolic Syndrome Incidence.” Your insights have been instrumental in refining our work.

 Following your request, we have revised the manuscript and addressed the concerns regarding plagiarism. The changes have reduced the plagiarism levels. Also, we have uploaded the requested FFQ file for your review.

We are hopeful that the revised manuscript now aligns with the standards for publication in the " PLOS ONE " Your interest in our research and the opportunity to enhance its quality are greatly appreciated. We remain open to any additional feedback or further revisions that you may recommend ensuring the manuscript meets the journal's criteria.

Once again, thank you for your commitment to advancing scientific discourse, and we look forward to your feedback on the revised version.

Best regards,

Parvin Mirmiran, Prof.

Nutrition and Endocrine Research Center, Research Institute for Endocrine Sciences, Shahid Beheshti University of Medical Sciences, Tehran, Iran

E-mail: parvin.mirmiran@sbmu.ac.ir

Parvin.mirmiran@gmail.com

---

## [Editor Report · Decision Letter 3]

12 Nov 2024

Associations between Spice or Pepper (Capsicum annuum) Consumption and Diabetes or Metabolic Syndrome Incidence

PONE-D-24-24401R3

Dear Dr. Hosseini-Esfahani,

We’re pleased to inform you that your manuscript has been judged scientifically suitable for publication and will be formally accepted for publication once it meets all outstanding technical requirements.

Kind regards,

Zohreh Sajadi Hezaveh

Academic Editor

PLOS ONE
---

## [Editor Report · Acceptance letter]

19 Nov 2024

PONE-D-24-24401R3 

PLOS ONE

Dear Dr. Hosseini-Esfahani, 

I'm pleased to inform you that your manuscript has been deemed suitable for publication in PLOS ONE. Congratulations! Your manuscript is now being handed over to our production team.

Kind regards, 

on behalf of

Dr. Zohreh Sajadi Hezaveh 

Academic Editor

PLOS ONE